# Compositional dependence of the fragility in metallic glass forming liquids

Sebastian A. Kube [1✉], Sungwoo Sohn[1], Rodrigo Ojeda-Mota[1], Theo Evers[1], William Polsky[1], Naijia Liu [1], Kevin Ryan[2], Sean Rinehart[3], Yong Sun[3] & Jan Schroers [1✉]

The viscosity and its temperature dependence, the fragility, are key properties of a liquid. A low fragility is believed to promote the formation of metallic glasses. Yet, the fragility remains poorly understood, since experimental data of its compositional dependence are scarce. Here, we introduce the film inflation method (FIM), which measures the fragility of metallic glass forming liquids across wide ranges of composition and glass-forming ability. We determine the fragility for 170 alloys ranging over 25 at.% in Mg–Cu–Y. Within this alloy system, large fragility variations are observed. Contrary to the general understanding, a low fragility does not correlate with high glass-forming ability here. We introduce crystallization complexity as an additional contribution, which can potentially become significant when modeling glass forming ability over many orders of magnitude.

[1] Department of Mechanical Engineering and Materials Science, Yale University, New Haven, CT, USA. [2] School of Engineering and Applied Science, Yale University, New Haven, CT, USA. [3] Department of Applied Physics, Yale University, New Haven, CT, USA. ✉email: Sebastian.Kube@aya.yale.edu; jan.schroers@yale.edu

The viscosity $\eta$ is the central property of a liquid[1,2]. It determines the macroscopic resistance to flow and governs dynamic processes such as diffusion[3,4] and structural relaxation[1,5]. With increasing temperature, thermal activation lowers the viscosity. In addition, the liquid's atomic equilibrium structure continuously adjusts to the increasing temperature, which further decreases the viscosity. The type of liquid (e.g., atomic, molecular, covalent network), its composition, and its atomic structure determine the magnitude of this structural contribution[2,6–10]. Thus, they are distinctively reflected in the resulting viscosity–temperature-dependence, which is a key property widely known as *liquid fragility*[6]. Qualitatively, liquids are classified as strong or fragile using the Angell plot (Fig. 1a). To quantify fragility, the fragility parameter $m$ is commonly used[7] (Eq. (1)). The fragility is a property of the liquid state, not of the glass state in which the atomic structure is frozen[1,11] (Fig. 1b).

$$m := \frac{\mathrm{d}\log_{10}\eta}{\mathrm{d}(T_g/T)}\bigg|_{T=T_g} \qquad (1)$$

Metallic glass forming liquids are especially interesting to study the fragility. Their structural simplicity, in particular, compared to polymeric liquids, allows for ubiquitous conclusions. At the same time, they provide a diverse ground for exploration, as compositions can vary continuously and typically feature multiple constituent elements with large differences in atomic size, energetic and geometric interactions[12–14]. This can lead to a wide range of liquid structures and properties. Most importantly, the glass forming ability (GFA) quantified through the critical cooling rate $R_C$ can range over many orders of magnitude[15]. Many metallic liquids exhibit low GFA with $R_C > 10^8$ K/s[15]. Others at specific compositions can be undercooled below $T_g$ without crystallizing even at $<10^3$ K/s, allowing for bulk glass formation[16,17]. In general, the fragility correlates with many liquid and glass properties, including relaxation, diffusion, and crystallization kinetics[18–22], mechanical properties[23] and elastic constants[24]. Most importantly, it has been widely suggested that

strong liquids are correlated with high GFA[11,18–21,25–32]. Technologically, the viscosity and fragility are important for processing[33–35], e.g., in casting, thermoplastic forming, annealing, and aging.

While the viscosity and fragility are of such fundamental importance, they are challenging to measure. They require a combination of techniques to cover the full viscosity range over 14 orders of magnitude[10,11,20,36,37]. Alternative methods can estimate the fragility without direct viscosity measurements[11,38], but their applicability and accuracy are often limited. To determine the fragility of metallic glass forming liquids, methods covering the high viscosity range around $T_g$ are better suited, where viscosity changes are most significant and crystallization times are longest. However, these techniques typically require bulk samples. This restricts their applicability, since the composition region of bulk glass formation is usually small within a given alloy system[30,33,39–41]. Due to these challenges, only a small number of reliable fragility data are available so far (comprehensive list in ref. [18]). These data are mostly limited to bulk glass formers, and usually cover only a single composition within each alloy system. The lack of fragility data covering wider composition ranges within a given alloy system has prohibited a systematic study of the structural origin of fragility and its interplay with other factors[18,19].

We propose that the structure-property relationships of metallic glass-forming liquids and their glasses are best revealed by comparing the fragilities of different alloy compositions within the same alloy system. Thereby, the chemical elements are held constant, whereas only the atomic and electronic structure of the liquid are varied by means of the composition. The considered composition space must span across multiple orders of magnitude in GFA to address various questions: How large are the fragility changes with composition? Are these changes smooth? Are there specific compositions, which exhibit unusually high or low fragilities, possibly arising from particularly favorable or unfavorable packing configurations? How does the fragility correlate with GFA?

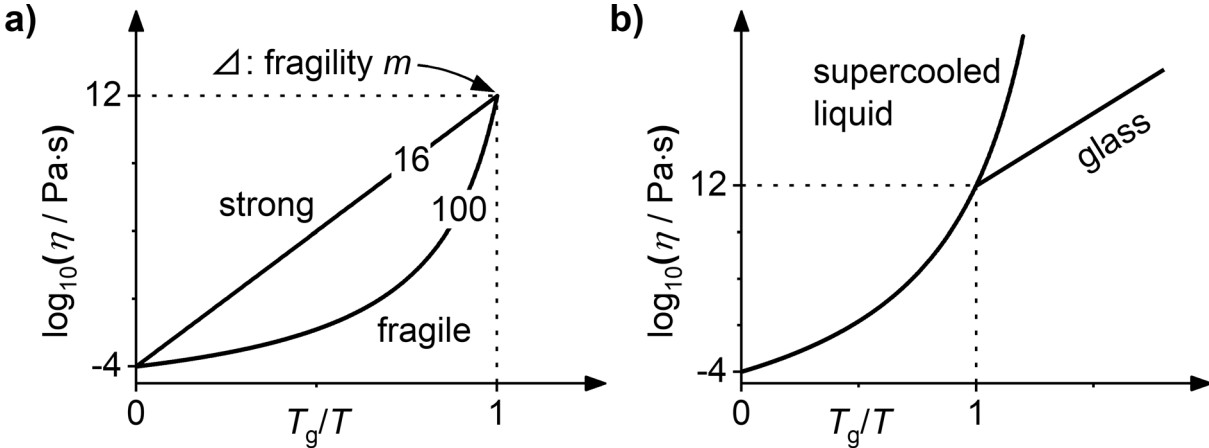

**Fig. 1 Viscosity-temperature dependence in liquids and glasses.** In Angell plots, $T_g/T$ is the inverse temperature scaled by the glass transition temperature $T_g$[6–9]. By rheological convention, $T_g$ corresponds to a viscosity of $10^{12}$ Pa·s, around which the calorimetric glass transition commonly occurs[6,11]. **a** Liquid fragility: In the strong limit, liquids exhibit a linear temperature dependence according to the Arrhenius Equation $\eta = \eta_0\exp\left(\frac{E_A}{k_B T}\right)$[1,7,8]. This arises exclusively from thermal activation and entails no structural changes, so the underlying flow mechanism and activation energy $E_A$ remain unchanged[1,7,8]. By comparison, fragile liquids exhibit lower viscosities throughout, and a steep increase upon approaching $T_g$ resulting from large structural changes[1,2,7,8,50]. Specifically, with decreasing temperature flow requires increasingly cooperative rearrangement, leading to growing activation barriers. The fragility parameter $m$ is the slope at $T_g$. It ranges from 16 for Arrhenius behavior, most closely realized by $SiO_2$[6], and exceeds 100 for fragile liquids[6,7]. **b** Glass transition: The supercooled liquid in metastable equilibrium can continuously adjust its structure to temperature changes. This is the origin of fragile, non-Arrhenius behavior. By contrast, the liquid falls from equilibrium into the glass state at $T_g$. Here, the time scale for structural changes becomes larger than the experimental time scale[1,8,37]. This frozen-in isostructural glass configuration exhibits a shallow viscosity–temperature dependence governed by Arrhenius-type thermal activation[1,20,37]. (For simplicity, identical $T_g$ and fictive temperature are assumed here.).

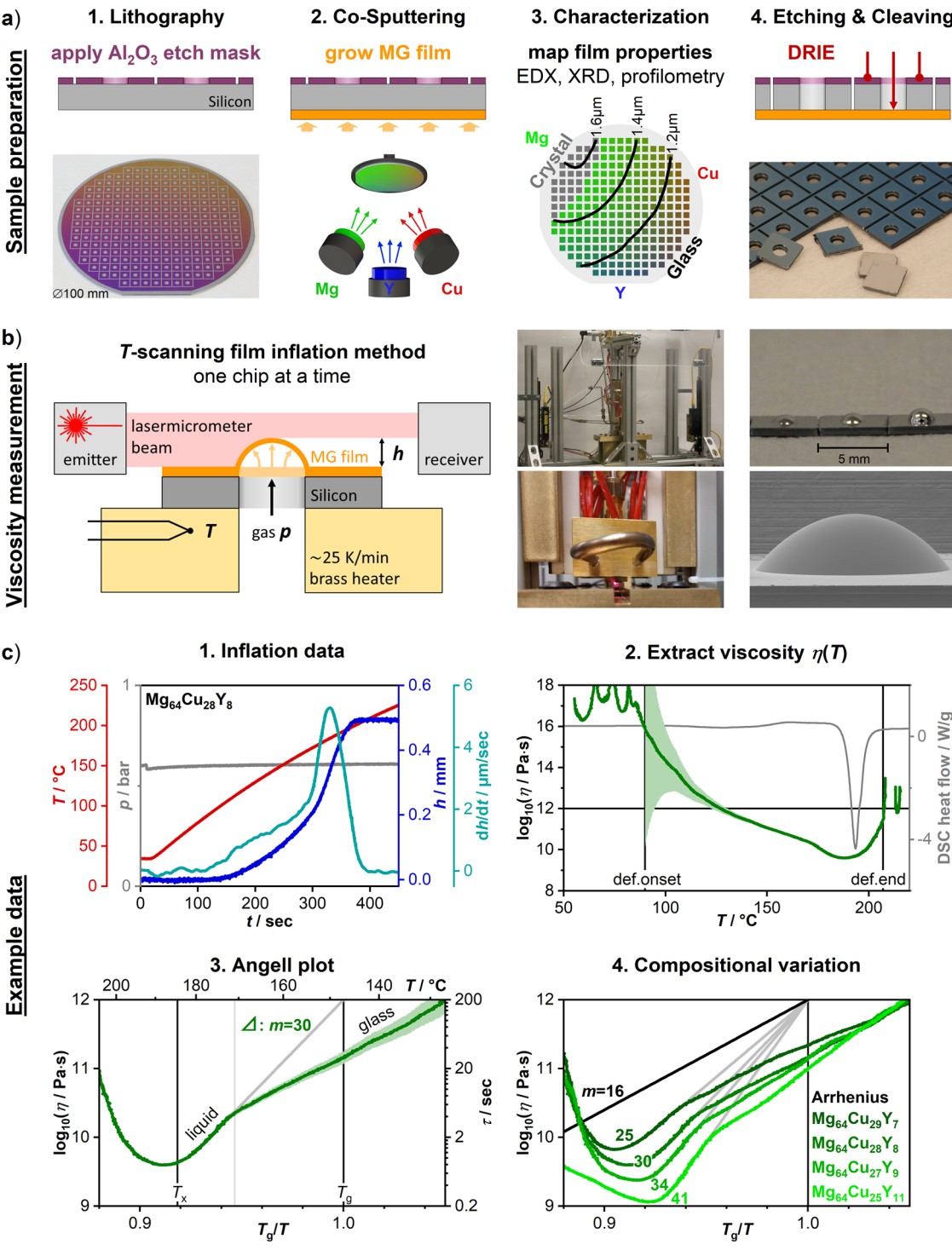

In this study, we lay the foundation for measuring the compositional dependence of the viscosity and fragility. We introduce our film inflation method (FIM), which relies on sputtering at a cooling rate of ~$10^8$ to $10^{10}$ K/s for glass formation[15,42], providing experimental access to glassy samples across wide composition ranges. We apply the film inflation principle[41] in temperature-scanning mode, implemented in our custom-built setup to precisely measure the temperature dependence of the viscosity. The sample fabrication and measurement process utilize high-throughput techniques and automation, combining efficiency with high quality standards[15,39,40,42–47]. Altogether, this enables

us to measure the fragility for an unparalleled 170 different alloys in the Mg–Cu–Y glass forming system[11,48,49].

## Results and discussion

**Film inflation method**. The FIM is illustrated in Fig. 2 (full details in "Methods" section and SI). In the sample preparation step, an alloy library is fabricated (Fig. 2a.2). This silicon wafer carries an amorphous Mg-Cu–Y film with a compositional gradient of 15–25 at.% variation in elemental concentrations[42,44]. After an etching step, the wafer is cleaved into 241 individual 5 mm square chips (Fig. 2a.4). Each chip differs in composition

**Fig. 2 Film inflation method (FIM). a** Sample preparation: (1) An $Al_2O_3$ etch-mask is applied onto a silicon substrate and defines 241 square chips, each with a 2 mm circular area of exposed silicon. (2) A compositionally graded Mg–Cu–Y film library is co-sputtered onto the opposite substrate side. Elemental concentrations vary by 15–25 at.%, neighboring chips differ by ~1 at.%. (3) High-throughput film characterization for each chip: EDX measures chemical composition (RGB representation), XRD identifies amorphous or partially crystalline film (colored vs. gray representation), profilometry determines film thickness (contour line representation). (4) Through deep reactive ion etching (DRIE), exposed silicon is vertically etched down to the film. Individual square chips are cleaved along the gridlines. Each chip features a 2 mm circular hole, across which the glassy film is freely suspended. **b** Viscosity measurement: Stress is applied to the free-standing film through gas pressure $p$. Governed by its viscosity $\eta$ at temperature $T$, the film deforms into a bubble of spherical geometry (see photograph and micrograph). The bubble's height h is recorded over time to calculate $\eta(T)$. In our automated setup, the silicon chip is pressed against a brass heater. $p$ and $T$ are recorded using a pressure gauge and thermocouple. The height h is recorded using a laser-micrometer, emitting a thin curtain of light, which the expanding bubble partially blocks. Operating in temperature-scanning mode (~25 K/min), the film reveals the viscosity–temperature dependence. **c** Example data: (1) Inflation data for $Mg_{64}Cu_{28}Y_8$ over time. (2) The resulting viscosity–temperature curve calculated using Eq. (2), along with the uncertainty estimate graph (green shaded graph). (3) Angell-plot: Two regimes are identified, the glass state and the metastable liquid state (compare Fig. 1b, and DSC trace in c.2). $T_g$ is determined by linear extrapolation from the liquid regime to $10^{12}$ Pa·s. $T_x$ is determined where the viscosity curve bends upwards. The fragility $m$ is determined as the slope extrapolated to $T_g$. (4) Using this approach, the fragility is measured at various compositions. While the example compositions here vary by only 4 at.%, the measured fragility exhibits a remarkably large variation from 25 to 41.

from neighboring chips by ~1 at.%, and therefore represents a glass of distinct chemistry and rheology. Such a fine compositional resolution is essential, as metallic glass characteristics can rapidly change over small composition ranges[41].

Each chip features a 2 mm circular hole at the center, across which the film is freely suspended. This free-standing film constitutes the actual samples, which is deformed in the viscosity measurement (Fig. 2b): One at a time, the chips are inserted into our measurement setup and heated at ~25 K/min. Simultaneously, a gas pressure is applied to the film. In response, the glassy film deforms through viscous flow and expands into a bubble. This deformation accelerates with increasing temperature and slows at the crystallization onset temperature $T_x$, after which the bubble reaches its final height as it crystallizes.

In this inflation test, the rate of deformation is a direct function of the viscosity at the given time and temperature. Conversely, we can infer the viscosity from the deformation rate. To this end, all relevant factors are recorded throughout the measurement: These include the pressure difference $p$, the height of the deforming bubble $h$, and the deformation rate $\dot{h} = \mathrm{d}h/\mathrm{d}t$. Further, the temperature $T$ is recorded, across which we scan to reveal the viscosity–temperature dependence. Before the inflation test, the initial film thickness $D_0$ is determined for each chip through profilometry. The base-radius $r_0 = 1$ mm is set by the selectively etched region. The viscosity $\eta(T)$ can then be calculated:[51]

$$\eta(T) = \frac{p}{24\,D_0} \cdot \frac{(r_0^2 + h^2)^3}{r_0^2 h^2} \cdot \frac{1}{h} \qquad (2)$$

Example measurement data are shown in Fig. 2c.1 for $Mg_{64}Cu_{28}Y_8$. The sample is heated from 35 to 225 °C with an applied gas pressure of 0.6 bar. At $t = 100$ s, the film's height grows detectably at an increasing rate and exhibits a notable surge in a second regime at 300 s. The deformation rate peaks at 330 s and subsequently drops rapidly as the film crystallizes.

The resulting viscosity graph is shown as a function of temperature (Fig. 2c.2) and inverse temperature $T_g/T$ in the Angell-plot (Fig. 2c.3). From the onset of detectable deformation, the viscosity continuously decreases upon heating. It reaches a minimum at the onset of crystallization, beyond which the apparent viscosity increases again as the material progressively crystallizes and finally solidifies. The measurement-error-based uncertainty estimation (green shaded graph around viscosity curve, see SI for details) reveals that we obtain high-confidence viscosity readings below $10^{12}$ Pa·s. Thus, our method can accurately probe a viscosity range of over three orders of magnitude, reaching below $10^9$ Pa·s.

The viscosity graph exhibits two distinct regimes prior to crystallization, which are more readily observed in the magnified view of the Angell-plot (Fig. 2c.3). The first regime at lower temperatures exhibits a shallow decrease of viscosity, whereas the second regime at higher temperatures exhibits a steep decrease. Comparing this to the schematic in Fig. 1b, we identify the first regime as the isostructural glass state, whereas the second regime corresponds to the state of the supercooled liquid in metastable equilibrium. To complete the analysis, we must thus focus on the second curve segment corresponding to the liquid state (Fig. 2c.3). $T_g$ can be determined by linear extrapolation to the conventional viscosity value of $10^{12}$ Pa·s, and the liquid fragility $m$ is determined from the slope (cf. Eq. (1)).

Using this approach, viscosity curves are measured at various compositions. For illustration, we present curves acquired for four different compositions along a line from $Mg_{64}Cu_{29}Y_7$ to $Mg_{64}Cu_{25}Y_{11}$ (Fig. 2c.4). These compositions vary over only 4 at.%. Nonetheless, the measured fragility significantly varies from 25 to 41.

**Validation of FIM**. In the following, our objective is to examine such compositional variation systematically through comprehensive composition maps. We will first examine the compositional dependence of $T_g$ and $T_x$, which allows us to validate our FIM experiment and data evaluation approach.

Both $T_g$ (Fig. 3a) and $T_x$ (Fig. 3c) exhibit a smooth and systematic compositional variation. The respective temperature values are low in the Mg-rich area and increase in the direction of higher Cu and Y concentrations. This behavior is qualitatively reasonable[52], as it correlates with the melting temperatures of the pure elements (see SFig. 3).

Next, we compare $T_g$ and $T_x$ to DSC-based literature values $T_{g,Lit}$ and $T_{x,Lit}$ of bulk and ribbon glasses reported in this composition region (see SI). For $T_g$ (Fig. 3b) we find good agreement, validating both the observed qualitative compositional variation as well as the absolute values determined, which suggests that FIM produces reliable data.

For $T_x$ (Fig. 3d), the FIM-based values are consistently ~10 °C lower than the literature values. This observation indicates that the film samples exhibit a systematically lower crystallization resistance than the bulk and ribbon glasses. This is reasonable, given that as-sputtered films often incorporate contaminants, particularly oxygen, as each atom is at some point exposed on the surface of the growing film. Further, such films exhibit higher volume fractions of interfaces and heterogeneities. Together these factors can promote heterogeneous nucleation, which makes films

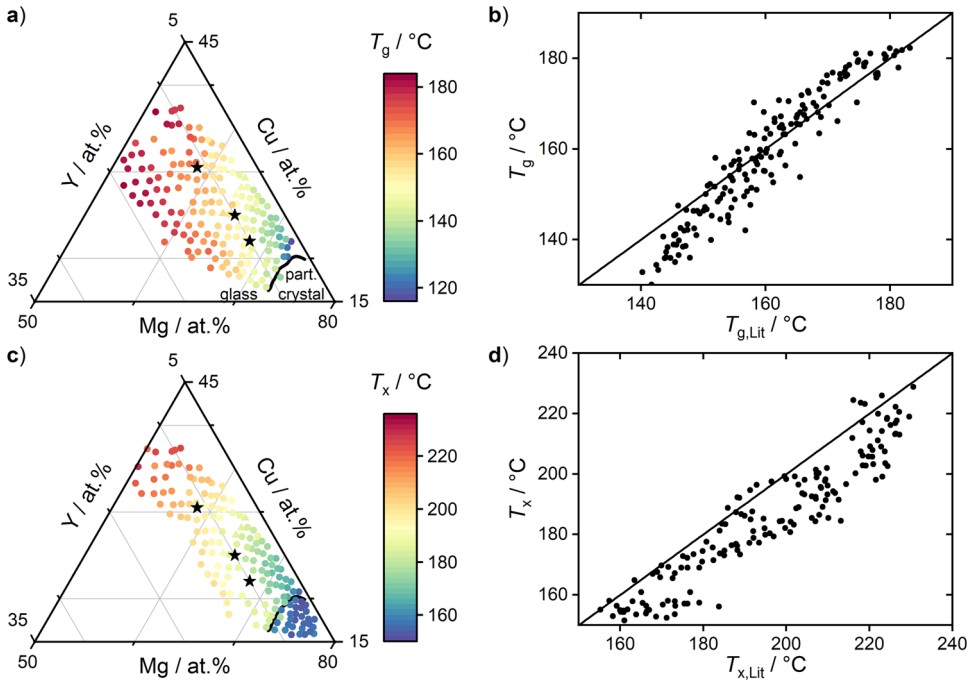

**Fig. 3 $T_g$ and $T_x$ in Mg–Cu–Y.** **a**, **c** Composition maps determined through FIM: Values vary smoothly and correlate with the pure element melting temperatures (see SFig. 3). **b**, **d** Validation by comparison with literature values $T_{g,Lit}$ and $T_{x,Lit}$: Our FIM values correlate strongly with DSC-based literature values, confirming that FIM yields qualitatively and quantitatively reliable data. $T_x$ values are systematically lower by ~10 °C, indicating a reduced crystallization resistance in the film. Note: In Figs. 3 and 4, black stars represent bulk-glass forming compositions. Triangle markers (near middle star) represent curves in Fig. 2.c.4. Sets of available composition map points can differ. For example, some points available at high Y concentrations in (**a**) are not available in (**c**). Here, the expanding films burst before reaching $T_x$.

more prone to crystallization. Notwithstanding this offset, the correlation between FIM data and literature data is strong.

**Fragility in the Mg–Cu–Y system.** With the confidence gained from validating FIM, we now use FIM to determine the fragility across the Mg–Cu–Y system (Fig. 4a). For $Mg_{65.6}Cu_{24.8}Y_{9.6}$ we observe $m = 44$. Within our data, this is the closest available composition to $Mg_{65}Cu_{25}Y_{10}$ (middle black star), the only alloy for which a fragility value of $m = 45$ has been reported in the literature[11,36].

Across the considered composition space, the fragility varies smoothly (Fig. 4a). Overall, $m$ ranges from 46 down to 16. The compositional variation is as high as $dm/dc = 10/at.\%$, which is the largest gradient reported within a metallic glass forming system to date[53–55].

Fragility variations of this magnitude are remarkable. Large fragility variations are commonly attributed to different materials classes, in particular, due to differences in interaction and structure type (e.g., covalent network in $SiO_2$ glass, metallic bonding in $Mg_{65}Cu_{25}Y_{10}$, intermolecular attraction in o-terphenyl). More recent evidence suggests that large fragility variations may also occur amongst metallic glasses, even within the same alloy system[53–55].

Our FIM fragility data presented here are the first to systematically resolve a wide composition region within a single ternary alloy system. Within this system, the type of bonding and the constituent elements remain unchanged. Therefore, the observed fragility variations must originate primarily from composition-dependent changes to the atomic and electronic structure, which presumably lead to different packing motifs at different compositions[56–59] and altered atomic interaction potentials[60,61]. This is particularly surprising for those alloys

with fragilities close to ~16, approaching Arrhenius behavior as previously only observed for $SiO_2$.

The here determined fragility-composition landscape also exhibits noticeable features of complex shape. For example, a ridge of approximately constant and high fragility stretches out along the line of $Mg_{70-x}Cu_{20+x}Y_{10}$ ($x$ from 0 to 12). Along the perpendicular direction, the fragility drops steeply. A second ridge, parallel to the first and separated by a valley, is present along the line of $Mg_{65-x}Cu_{18+x}Y_{17}$ ($x$ from 0 to 10). These ridges may originate from distinct packing motifs which dominate along such lines.

Indeed, it has been widely argued that small changes in composition can significantly alter atomic packing and dramatically affect the fragility and GFA[10,56–59]. Comparing predictions from proposed models of efficient cluster packing[59] to the fragility landscape measured here does not show qualitative agreement. This discrepancy likely originates from oversimplified model assumptions, which typically include hard-sphere atoms, neglect bond directionality, and ignore possible ordering from preferential chemical interaction. For Mg–Cu–Y this representation is certainly not accurate (e.g., reduced Mg–Cu bond length, HCP geometry for Mg and Y, FCC for Cu, exothermic Cu–Y interaction).

**Fragility and glass forming ability.** Our data also allow conclusions about the relationship between fragility and GFA. It has been widely suggested that a high GFA correlates with strong liquid behavior[11,18–21,25–32]. Strong liquids exhibit higher viscosities across the vitrification temperature range, which slows the crystallization kinetics and thus reduces the critical cooling rate $R_C$ required for glass formation. In addition, strong liquids are thought to be more densely packed and hence exhibit an enthalpy

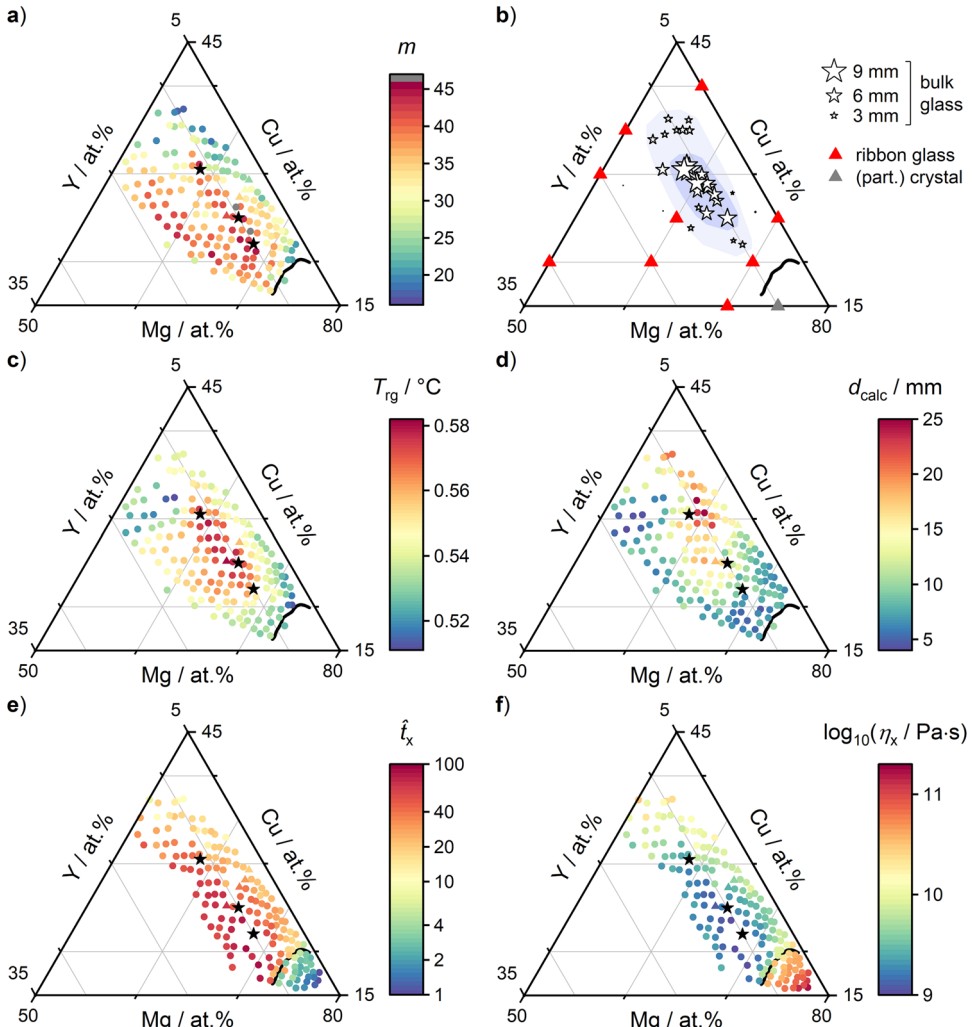

**Fig. 4 Composition maps of the fragility and other properties, and comparison to glass forming ability in Mg–Cu–Y. a** Variation of fragility $m$ as determined via FIM. **b** Curated map of glass forming ability (GFA). Bulk glasses with high GFA can be formed in a narrow region along $Mg_{68-x}Cu_{22+x}Y_{10}$ ($x$ from 0 to 13). GFA rapidly decreases to the sides. (Curated from published bulk and ribbon data, and additional self-prepared suction-cast wedge-type bulk samples. Full data and details in SI.) **c** The reduced glass transition temperature $T_{rg}$ (Eq. (3)) is calculated using $T_g$ data from FIM (Fig. 3a) and literature-based liquidus data (see SI). **d** Predicted critical casting thickness $d_{calc}$ calculated according to the model by Johnson et al.[18], which combines $m$ and $T_{rg}$ (Eq. (4)). **e** $\hat{t}_x$, the crystallization time in relaxation time units as an indicator of crystallization complexity (Eq. 5). For example, an alloy with $\hat{t}_x = 100$ requires 100 times the relaxation time to crystallize, which reflects the kinetic complexity of the crystallization process. **f** Viscosity at the crystallization onset $\eta_x$, which is the lowest viscosity value reached before crystallization. A higher crystallization complexity allows to reach lower $\eta_x$, even for those compositions which exhibit the highest fragility values.

closer to the competing crystal phases, which reduces the driving force of crystallization[20].

While intuitively reasonable, examples which do not follow this correlation have been reported[18,22,62,63]. Most notably, $Pd_{42.5}Cu_{30}Ni_{7.5}P_{20}$ is considered the glass former with the highest known critical casting thickness and equivalently the lowest critical cooling rate, but it exhibits a comparatively high fragility of $m = 58$[18,22]. Conversely, $Zr_{80}Pt_{20}$ appears to exhibit a low fragility, but does not form a glass even in melt-spun ribbons[62]. The structural origin of such exceptions is under active investigation[63,64].

To test this correlation, we present a map of GFA in Mg–Cu–Y (Fig. 4b). While GFA is sufficient for the formation of ribbon glasses ($R_C \leq 10^6$ K/s) across most of the composition range, bulk glass formation ($R_C \leq 10^3$ K/s) is only found in a narrow region along the line $Mg_{68-x}Cu_{22+x}Y_{10}$ ($x$ from 0 to 13).

When comparing the fragility and GFA maps (Fig. 4a, b), the area of highest GFA coincides with the ridge of highest fragility values. This is remarkable, since according to the current

understanding the opposite should be the case; high GFA should be associated with a low fragility[11,18–21,25–32]. Evidently, this correlation does not have to be strictly fulfilled. Other contributions besides the fragility must determine GFA.

Indeed, previous studies argue that the combination of $T_{rg}$ and $m$ must be considered[18,19,32]. In particular, by combining both parameters into regression models, Johnson et al.[18] successfully modeled GFA for a comprehensive collection of BMGs. Similarly, Greer et al.[19] modeled congruent crystal growth rates for various metallic and non-metallic liquids.

$$T_{rg} = \frac{T_g}{T_L} \quad (3)$$

Here, $T_{rg}$ is the reduced glass transition temperature normalized by the liquidus temperature $T_L$, as originally introduced by Turnbull[65]. The temperature interval from $T_L$ to $T_g$, across which the liquid must be cooled, decreases as $T_{rg}$ approaches unity.

Further, the maximum driving force and velocity of crystallization become smaller[65], which promotes vitrification.

Together, $T_{rg}$ and the fragility $m$ are complementary contributions: While $T_{rg}$ quantifies the width of the temperature interval of vitrification, $m$ quantifies how quickly liquid kinetics slow down throughout this interval.

To test this approach on our data, we first examine $T_{rg}$ alone. Our $T_g$ data (Fig. 3a) are combined with $T_L$ data compiled from the literature (see SFig. 4[49,66],) to generate the $T_{rg}$ map shown in Fig. 4c. We find that the region of high $T_{rg}$ coincides well with the region of high GFA. This is primarily due to a deep eutectic located at $Mg_{65}Cu_{25}Y_{10}$[49]. From this perspective, $T_{rg}$ appears to be the dominant contribution to GFA within this alloy system.

We then combine $T_{rg}$ and $m$ according to the two models. The results are qualitatively equivalent, and we show only the results according to Johnson et al. here (see SFig. 6 for Greer et al.). In Johnson's model, the critical casting thickness $d_{calc}$, is predicted as[18]:

$$\log_{10}(d_{calc}) = 12.8 \cdot T_{rg} - 0.02405 \cdot m - 5.18 \quad (4)$$

Figure 4d shows the resulting composition map. At $Mg_{65}Cu_{25}Y_{10}$ we find approximate agreement (here: 11.6 mm, Johnson[18]: 8.8 mm, experimental[48]: 7 mm), suggesting potentially useful GFA predictions. However, over the wider composition range the agreement is not good. Even far from the bulk forming region, $d_{calc}$ values of 5 mm or higher are predicted. This discrepancy is particularly obvious on the boundary line of crystallization under sputtering conditions. Here, the critical cooling rate is ~$10^8$ K/s, which corresponds to a critical casting thickness of <1 μm[15,67]. Yet, for $Mg_{72}Cu_{19}Y_9$ ($m = 31.8$, $T_{rg} = 0.53$) a $d_{calc}$ value of 7 mm is predicted, which is four orders of magnitude larger. This discrepancy cannot result from hypothetical errors in the determined fragility since Eq. (4) would require $m = 190$ for 1 μm.

The qualitative agreement is also poor. $d_{calc}$ does not match the narrow, long contour of the high GFA region. Further, at $Mg_{60}Cu_{30}Y_{10}$ a GFA as high as $d_{calc} = 25$ mm is predicted, resulting from low fragility values $m \approx 25$ overlapping with high $T_{rg}$ values $\geq 0.56$. The experimental GFA data do not support this prediction.

Overall, the above observations suggest that the fragility is not the dominant contribution to GFA in Mg–Cu–Y. The quantitative models[18,19] seem to overestimate the contribution of $m$.

We argue that Johnson's model is effective when applied to bulk metallic glasses with high GFA[18]. However, $T_{rg}$ and $m$ alone are insufficient to explain GFA over the full GFA spectrum. Our Mg–Cu–Y data cover a broad composition region, in which GFA ranges from poor glass formers to BMG formers. Thus, additional features can emerge as decisive contributions to GFA.

**Glass forming ability beyond $T_{rg}$ and $m$.** In general, the processes determining GFA are intricate. In addition to $T_{rg}$ and $m$, the crystallization pathway must be considered. We suggest describing this additional contribution as *crystallization complexity*.

The crystallization of bulk glass forming alloys is very complex[12,16,68]. It requires the formation of multiple phases at different compositions through time-consuming diffusion processes, leading to long crystallization times and high GFA. Marginal glasses exhibit a lower crystallization complexity[15]. Finally, pure elements and solid-solution-forming alloys crystallize along simple polymorphic pathways, where a single crystalline phase directly forms at the same composition without prior demixing[15,19,30,42,69]. This requires only short-range topological rearrangement, and hence occurs much faster than nonpolymorphic crystallization in BMGs.

A high crystallization complexity is required for high GFA, but it is not captured by $T_{rg}$ or $m$. $T_{rg}$ quantifies the width of the supercooled liquid interval and $m$ quantifies the kinetic slowdown throughout. But neither parameter quantifies the inherent time required for crystallization.

Successful GFA models will require a broadly applicable crystallization complexity parameter. So far, TTT-diagrams quantify crystallization kinetics through isothermal crystallization times $t_x(T)$[16,20,70]. Close to $T_g$, growth governs $t_x(T)$ [16] with a temperature dependence dominated by the viscosity. To eliminate this alloy-specific temperature dependence, we propose to normalize $t_x(T)$ by the relaxation time $\tau(T)$:

$$\hat{t}_x(T) = \left.\frac{t_x(T)}{\tau(T)}\right|_{T \gtrsim T_g} \quad (5)$$

$\tau$ quantifies the time scale required for structural rearrangement, in particular for reaching metastable equilibrium at a given temperature, and it, therefore, provides a readily available reference time scale for crystallization. Generally, $\tau(T)$ is proportional to $\eta(T)$. Based on empirical observations, a relaxation time of ~200 s is typical of the calorimetric glass transition and corresponds to a viscosity value of $10^{12}$ Pa·s, which by established convention is the assumption made when direct relaxation time measurements are not available[6–9,11,22].

By normalization, the $\hat{t}_x$ indicator then independently represents crystallization complexity and allows to compare alloys directly (see SFig. 7). It is an approximate indicator, because elemental diffusivities and crystal growth rates can decouple from the viscosity close to $T_g$[19,71,72]. Nonetheless, metallic glasses display a strong correlation between $t_x(T)$ and $\tau(T)$, which reasonably matches proportionality close to $T_g$[35,73].

Most importantly, $\hat{t}_x$ provides a meaningful representation of crystallization times: A liquid crystallizing within just one relaxation time unit is expected to follow a direct transformation path typical of polymorphic crystallization[15]. A liquid requiring many relaxation time units undergoes more complex crystallization, as observed for the best glass forming alloys[35].

For illustration, we use a modified version of Eq. (5) to determine $\hat{t}_x$ values from our FIM curves. Figure 4e shows that $\hat{t}_x$ varies widely and correlates with GFA (details in SI).

Altogether, a high crystallization complexity is found in alloys with high GFA. Its contribution may not always be obvious, in particular when comparing bulk metallic glasses, which generally exhibit a high crystallization complexity. Nevertheless, our results suggest that crystallization complexity becomes decisive when comparing alloys across many orders of magnitude of GFA.

In summary, our FIM technique enables studying the composition and temperature dependence of the viscosity over wide composition ranges. For the Mg–Cu–Y alloy system, FIM reveals a remarkably large range of fragility variation, with dramatic changes occurring even over small composition ranges. Since the constituent elements remain the same, these large fragility changes must originate primarily from composition-dependent changes to the atomic and electronic structure.

Further, our data show that a low fragility does not necessarily correlate with high GFA. The models based on this correlation seem to overestimate the contribution of fragility to GFA. In addition to the previously suggested parameter combination of $T_{rg}$ and $m$, we propose that a high crystallization complexity contributes to GFA and becomes the dominant contribution when alloys are compared over many orders of magnitude of GFA.

## Methods
Full method details are provided in the SI.

**Sample preparation and characterization**. The patterned $Al_2O_3$ mask is applied using photolithography and e-beam evaporation. The Mg–Cu–Y film is grown onto the opposite substrate side. $Mg_{65}Cu_{25}Y_{10}$ is targeted as center-composition, which corresponds to the bulk glass most studied in this alloy system. Combinatorial co-sputtering produces a compositional gradient. For example, the film exhibits higher copper concentrations closer to the copper sputtering source, as illustrated in red (Fig. 2.a.2). High-throughput characterization is conducted based on automated XY-stages, providing local film measurements for each chip; energy-dispersive X-ray spectroscopy (EDX) for composition, X-ray diffraction (XRD) for amorphicity, profilometry for the film thickness needed to compute the viscosity in Eq. (2). After deep reactive ion etching (DRIE), the individual square chips are cleaved off the wafer.

**Viscosity measurement**. In our experimental setup, the silicon sample chip is firmly pressed against a brass heater, creating a gas-tight seal and good thermal contact. The pressure difference is recorded using a pressure gauge, and the temperature is recorded using a thermocouple within the heater at close proximity to the sample. Inert argon gas is used to apply the gas pressure. The evolution of the bubble height is recorded using a calibrated laser micrometer (Micro-Epsilon ODC1200/90-2, 2 mm detection range): The emitter projects a thin curtain of light, which is carefully aligned onto the chip surface and into the bubble's center plane. As the bubble grows, it continuously blocks more of this light from arriving at the receiver. The receiver in turn converts the reduced light signal into a height reading. Our automated setup completes one sample measurement within 20 min. The expansion data are processed to extract the viscosity, corresponding uncertainty estimate, fragility, and characteristic temperatures using Python. For a number of samples measured, the bubbles burst before reaching crystallization. The analysis accounts for such cases and data are provided accordingly. The gray markers in Fig. 4a represent compositions with $m$ determined at >46. However, these samples burst early and the obtained fragility values are doubtful.

## Data availability

All data including the FIM data generated in this study are provided in the Supplementary Information file. Source data are provided with this paper.

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

## Acknowledgements

Financial support from the National Science Foundation through NSF-DMR (grant no. 2104316) is gratefully acknowledged. The authors thank Nicholas Bingham, Min Li, Michael Rooks, Kelly Woods and Amit Datye for characterization assistance, Vincent Bernardo and Nicholas Bernardo for machining assistance, and Punnathat Bordee-nithikasem, Luis Perez Lorenzo, Aya Nawano and Rodrigo Reboucas for insightful discussions.

## Author contributions

S.A.K. and J.S. designed this study. S.A.K., S.R., and Y.S. designed the cleanroom fabrication process. S.A.K., R.O.M., T.E., W.P., N.L., and K.R. designed and built the automated FIM measurement setup. S.A.K. fabricated the FIM library and resulting sample chips and conducted all measurements. S.A.K. and S.S. fabricated and characterized the complementary bulk samples. S.A.K. and J.S. analyzed the results and wrote the manuscript.

## Competing interests

The authors declare no competing interests.
