## [Peer Review File · Nature Communications]

Title: Compositional dependence of the fragility in metallic glass forming liquidsREVIEWER COMMENTS

Reviewer #1 (Remarks to the Author):

This is an excellent and exciting work, which definitely deserves publication in Nature Communications. In this work the authors determine the fragility of glass forming liquid over wide ranges of composition, by developing the film inflation method. Fragility is a very important property of a liquid which relates to many physical properties. However, it is not easy to evaluate fragility accurately by experiment. Often kinetic fragility measured by thermal measurement by DSC is used as a substitute, but thermal fragility often disagrees with kinetic fragility. The results shown here are very impressive and surprising. For instance, they find that the fragility varies quite strongly with composition, which has not been expected. Also, at least for the composition (Mg-Cu-Y) the results contradict the widely held belief that a strong liquid is a good glass former. Instead, they find better agreement with the old T_g/T_l ratio method by Turnbull. It is clear that this work opens up a new window in liquid research, and further work would greatly expand the horizon of our knowledge.

However, I think some revision will increase the value of the paper. The experimental part (Section 1 – 4) is excellent, but the analysis in Section 5 is too long and less convincing. For instance, the discussion on crystallization complexity (lines 379 – 430) is half-baked and not directly connected to the results shown here. Actually, the newly defined timescale, eq. (7), is not very successful. This section considerably weakens the manuscript. I suggest either to move this section to Supplement, or to publish it separately with deeper analysis.

Reviewer #2 (Remarks to the Author):

This manuscript reports an extensive investigation of glass formation in a large number of Mg-Cu-Y alloys. In addition, viscosity and fragility have also been measured. It is true that reliable fragility data for the metallic glasses are confined to limited compositions. The combinatorial technique has proven to be very efficient in studying a large number of alloys. This technique, along with the use of the film inflation method for measuring viscosity, made such extensive studies possible.

I agree with the main conclusion that Johnson's work, which observed an empirical relation between critical thickness, fragility and T_{rg} , is an approximation; other factors are also involved. The general understanding is that fragility, among many other factors, determines glass formability (GFA). The authors have cited Pd-based glasses as one such example. OTP, one of the most fragile liquids known, is hard to crystallize. On the opposite end, Zr₈₀Pt₂₀, which is a strong liquid (even stronger than Vit 106, J. Mater. Res. 32(2017)2638), does not form a glass even by the melt-quench technique. Such questions have been raised in a number of publications (JNCS 525(2019)119673,) and structural origins for this discrepancy have been discussed (J. Phys.: Condens. Matter 29 (2017) 023002).

Although the TTT diagram has been extensively used to study GFA, use of the normalized time for crystallization, t_x , with respect to the relaxation time is a new approach. However, I do not understand

why it works. First, estimating the relaxation time from the viscosity data is problematic. The relaxation time should be determined by the diffusion constant. There are two relaxation processes dominant in liquids near T_g , the fast (Alfa) and the slow (Beta) ones. It has been shown that the diffusion rate is related to the Beta relaxation (PRL 109(2012)095508). In contrast, the viscosity is mainly determined by the fast process (Alfa relaxation). It was recognized long ago (Acta Metall. 37(1989)1355) that the two do not necessarily correlate. Many studies of metallic liquids have also shown breakdown of the Stokes-Einstein relation in metallic liquids far above their T_g s. Therefore, I do not know how far this estimate is reliable.

There is another difficulty. The time to crystallize depends both on the nucleation and growth rates. The growth rate is determined predominantly by the diffusion constant. Even if I accept the relaxation time from the viscosity data as a crude approximation, then the normalized quantity, \hat{t}_x , takes care of the diffusion part; nucleation contribution is completely ignored. Does it mean that either nucleation is unimportant or it is identical for all alloys studied? I find this difficult to accept.

It is not clear how this modified quantity, \hat{t}_x , is better than the bare time for crystallization, t_x , as a measure of GFA. How about showing such a comparison?

The manuscript is well written. The results are new. However, many questions, other than raised above, remain open.

1. Any effect of substrate, oxidation on viscosity measurements? It is well known that Y and Mg are susceptible to oxidation. Even the highest purity inert gases contain many tens of ppm oxygen. Thin films make them more susceptible to oxidation, as has been observed in the T_x measurements (lines 240-245) and Figs. 3(b) and 3(d). Have the authors checked how the measurements change after repeated thermal cycling of the same sample without crossing the crystallization temperature? Any effect of the heating rate?
2. The statement that large fragility variation is expected only in different material classes (lines 271-272 and 434-435) is inaccurate. It was known from studies of a binary system, such as CuZr (J. Phys.: Conf. Ser. 144 (2009)012094, JNCS 376(2013)205).
3. Lines 134-135 and 273. Types of bonding may not change, but the interaction potential definitely changes with composition even in a given system. The heats of mixing provide direct evidence for that. It is a common experience that an interaction potential developed for a particular alloy composition does not work well in the MD simulation studies even when the composition is changed by a few atomic percents. Such changes with the interaction potential are expected to change the fragility (PNAS 112(2015)13762), J. Stat. Mech. 084001(2016), J. Chem. Phys. 135(2011)194503).
4. Lines 124-125: The statement that the indirect methods described in ref. 17 and 50 do not require "bulk samples" is incorrect. I suppose the authors meant "bulk metallic glasses". For the determination of fragility from specific heat measurements, a better ref. is J. Chem. Phys. 125(2006)074505, since it covers a wide variety of materials.
5. Lines 274-275, the authors provide no experimental evidence that the fragility change is due to structural changes. This speculation is probably based on literature data. Proper references should be included.

Reviewer #3 (Remarks to the Author):

In the manuscript, the authors present a new technique for measuring the composition and temperature dependence of the viscosity over wide composition ranges. Based on these data associate with the GFA information, they propose that a low fragility does not necessarily correlate with high glass forming ability. However, it is not a surprised conclusion which has been widely accepted. They also tested the predictability of some GFA models, however, they gave only phenomenological descriptions of their conclusions. Thus, I think it is not suitable for Nature Communications.

1. Figure 1b is not a strictly correct schematic plot describing the relationship between the viscosity properties of glass and supercooled liquid. According to the PEL concept, when the temperature falls below T_g or other critical temperature, the system is trapped in a meta-basin. Thus, the individual glass state is just a projection of the supercooled state. If the measurement time or wait time is long enough, the relaxation behavior of the so-called glass and the supercooled liquid should be the same even if the viscosity is below 1012 Pa.s or other values.
2. The authors conclude that "in contrast, the here reported fragility variations result exclusively from changes to the atomic structure as the composition changes". However, till now, we have not really understood how atomic structure changes with composition. The EAM type interaction is not precise enough to understand the minor-element effects, and the "atomic-cluster" model is still not good enough to describe the SRO and MRO of amorphous structure.
3. In this measurement, they have more than 200 data. I wonder if they can come up with a reasonable GFA-prediction model based on machine-learning or other AI methods. It should be better than just give some phenomenological descriptions of the influences of different parameters.
4. To be honest, this technique is a useful one.

Reviewer Response, Submission NCOMMS-22-00021A (second revisions)

“Compositional dependence of the fragility in metallic glass forming liquids”

Sebastian A. Kube and Jan Schroers, et al.

We thank the reviewers for carefully reading the manuscript, and for providing constructive comments and suggestions to improve the manuscript. Based on these comments, we have fundamentally revised the relevant sections.

*Please find below in **blue** our detailed, point-by-point responses to the reviewers’ comments. Also, we provide supporting segments from the revised manuscript in **light blue italic**. All changes have been highlighted in blue font in the revised manuscript.*

Reviewer #1:

The authors adequately addressed the concerns described earlier and significantly modified the text. I recommend publication as revised.

We thank the reviewer for this assessment.

Reviewer #2:

As I stated in my earlier report, this is an extensive investigation that produced a huge amount of experimental data which will be useful for researchers in this field. There were a few open questions that myself and other referees raised. The authors have tried to answer those questions as far as possible and revised the manuscript accordingly. Aside from the importance of the new data, the authors have proposed a new metric to assess glass formability. Although not an open and shut case, the new metric may, to some extent, quantify the complexity of crystal phase/phases formation. It is understood by many active researchers in this field that complex pathways from glass/supercooled liquid to crystal are important factors for glass formation; whether that complexity comes from the structural difference between the liquid and crystal phases is an open question. I am not aware of any previous work that tried to quantify this complexity. As a first attempt in this direction this work is important and merits publication. Time will tell how useful this metric will be!

We thank the reviewer for this assessment and agree.

The raw data that came out from this investigation will be of interest to researchers who are trying to develop better theories for glass formation. The authors have provided some links in the supplement sections where such data will be available. I think they should also provide such data for the fragility parameter, in case I missed such a link.

We thank the reviewer for pointing this out. We agree that the reader should be able to find these valuable data easily. Indeed, the fragility data are already enclosed in the "SupplementaryFiles_ResearchData" folder, which we uploaded upon submission. To help the reader to quickly locate the fragility data, we have added the following statement to the Supplementary Contents section as recommended by the reviewer:

All FIM data are enclosed in "FIMData.xlsx", including the fragility data.

Reviewer #3:

I had carefully read the revised manuscript and the response to the comments. Some improvement can be found, but there is no substantial improvement.

We thank the reviewer for the time and effort taken to assess our revised manuscript.

However, we are disappointed by the reviewer's assessment as it shares neither our own excitement for this research, nor the support and approval we receive from the other two reviewers, nor the enthusiastic interest we have received from the metallic glass community when we presented these results at two recent conference meetings and two seminars.

We want to reiterate that we have already addressed all of the reviewer's comments in great detail in the first review round. It appears that the reviewer is actually satisfied with the majority of these revisions, as these aspects are no longer mentioned in this second assessment.

In particular the resolved aspects we addressed include:

- Evaluating the predictability of literature GFA models.
- Justification of Figure 1.
- Connection between fragility and structure
- Why we decided against using machine learning here.

Since these comments have been resolved, one can conclude that the manuscript has in fact improved substantially.

In the first review round we also clarified why our findings are novel and of significant interest to the community. The other two reviewers are also excited about these findings. Yet, reviewer #3 does still not seem to be satisfied. We will try to address this through additional clarification.

The reason why I think this paper is a result based on phenomenon description is that it does not give any characterization of the correlation between parameters.

Our manuscript provides extensive quantitative information on T_g , T_x , T_L , GFA, m , T_{rg} , d_{calc} , \hat{t}_x , and η_x . Each quantity is visualized as a function of composition over a wide range. This is a comprehensive quantitative assessment. Thanks to our FIM data, this assessment goes far beyond previous published analyses.

To address the reviewer's concerns "that no characterization of the correlation between parameters is given", we emphasize below that an extensive characterization of such correlations has in fact been carried out. Most importantly:

a) We characterize the correlation between m and GFA:

When comparing the fragility and GFA maps (Figure 4a and b), the area of highest GFA coincides with the ridge of highest fragility values. This is remarkable, since according to the current understanding the opposite should be the case; high GFA should be associated with a low fragility 14, 17, 18, 19, 21, 29, 30, 33, 34, 35, 36, 37, 38. Evidently, this correlation does not have to be strictly fulfilled.

b) We characterize the correlation between T_{rg} and GFA:

We find that the region of high T_{rg} coincides well with the region of high GFA. This is primarily due to a deep eutectic located at $Mg_{65}Cu_{25}Y_{10}$ ⁵⁶. From this perspective, T_{rg} appears to be the dominant contribution to GFA within this alloy system.

c) We evaluate the performance of d_{calc} , and thereby characterize how the previously established model combination of m and T_{rg} correlates with GFA:

At $Mg_{65}Cu_{25}Y_{10}$ we find approximate agreement (here: 11.6 mm, Johnson 17: 8.8 mm, experimental 55: 7 mm), suggesting potentially useful GFA predictions. However, over the wider composition range the agreement is not good. Even far from the bulk forming region, d_{calc} values of 5 mm or higher are predicted. This discrepancy is particularly obvious on the boundary line of crystallization under sputtering conditions. Here, the critical cooling rate is $\sim 10^8$ K/s, which corresponds to a critical casting thickness of $< 1 \mu m$ ^{26, 70}. Yet, for $Mg_{72}Cu_{19}Y_9$ ($m = 31.8$, $T_{rg} = 0.53$) a d_{calc} value of 7 mm is predicted, which is four orders of magnitude larger. This discrepancy cannot result from hypothetical errors in the determined fragility since Eq. (6) would require $m = 190$ for $1 \mu m$.

The qualitative agreement is also poor. d_{calc} does not match the narrow, long contour of the high GFA region. Further, at $Mg_{60}Cu_{30}Y_{10}$ a GFA as high as $d_{calc} = 25$ mm is predicted, resulting from low fragility values $m \approx 25$ overlapping with high T_{rg} values ≥ 0.56 . The experimental GFA data do not support this prediction.

d) We characterize the correlation between \hat{t}_x and GFA:

Error! Reference source not found. e shows that \hat{t}_x varies widely and correlates with GFA (details in SI).

e) We characterize the correlation between \hat{t}_x and η_x :

A higher crystallization complexity allows to reach lower η_x , even for those compositions which exhibit the highest fragility values.

f) We show and describe many additional correlations throughout the manuscript and in the Supplementary Information.

We can't get any quantitative information from Fig 4, but can only roughly judge whether the correlation is good or bad through the eyes.

We respectfully disagree.

Every single panel in Figure 4 provides extensive quantitative information: The position of each marker unambiguously encodes the composition of the corresponding alloy. The color of each marker unambiguously encodes the value of the corresponding quantity. The colorbars link the marker color to the quantity value. Thus, each of these colormaps provides quantitative information throughout.

The colormaps in Fig. 4 provide a strong visual representation of our data. They allow the reader to easily grasp and compare the composition dependency of each quantity. The reader can easily recognize correlations and discrepancies between all quantities.

I agree that this work provides a useful new method for measuring dynamic behaviors of glass forming liquids. Based on this experimental technique, the authors measured a number of data within one ternary alloy system, but it seems to me that this is a waste of the value of these data and thus loses the point of high-throughput research.

We respectfully disagree with the reviewer's statement. The value of these data is not wasted. Within this manuscript, these data unfold value in three ways:

- 1) Firstly, these data clearly demonstrate the robustness of our new FIM method.
- 2) Secondly, we have gleaned a number of significant insights from these data. These are:
 - The fragility varies over a remarkably wide range in Mg-Cu-Y. Large fragility variations have commonly been attributed to different materials classes, due to the fundamental differences in interaction and structure type. More recent evidence suggests that large fragility variations may also occur amongst metallic glasses, even within the same alloy system. Here, in Mg-Cu-Y we observe the largest change of fragility with composition that has been reported within a metallic glass forming system to date. The observed fragility variations must originate primarily from composition-dependent changes to the atomic and electronic structure, which presumably lead to different packing motifs at different compositions and altered atomic interaction potentials.
 - We find that a low fragility does not necessarily correlate with a high glass forming ability. Moreover, even the latest established and widely accepted literature models, which are based on the two correlations that a small fragility and a large T_{rg} lead to good GFA, clearly overestimate the contribution of the fragility in Mg-Cu-Y.

- Our data are the first to represent the wider spectrum of metallic glass forming alloys. Under these circumstances the previous models do not successfully predict GFA and thus do not offer useful predictions across Mg-Cu-Y. We find that these models are incomplete. Improved models must incorporate additional contributions to represent the broad spectrum of metallic glass forming alloys.
 - We identify the crystallization complexity as one such contribution. Our data suggest that this contribution becomes decisive when comparing alloys across many orders of magnitude of GFA.
 - To quantify this contribution for the first time, we propose the \hat{t}_x indicator. This indicator offers an approximation which in the long term will certainly be assessed and refined by the community. Within the manuscript, we explain that this parameter offers very reasonable and insightful estimates which capture the essence of crystallization complexity. Our data correlate well with GFA and support this proposal.
- 3) Thirdly, the above insights form the foundation to incorporate the concept of crystallization complexity into models of glass formation. The data show that the previous understanding of fragility and glass forming ability is insufficient and this additional contribution of crystallization complexity must be considered.

For the wider metallic glass community these data also unfold value: We provide all data in a clean format to the reader. This means that these data will be permanently and openly available to the entire community. We anticipate that other researchers will use these data to identify and test new correlations and build better models. This is most certainly in the spirit of modern high-throughput research.

We believe, and reviewers #1 and #2 share this view, that in the longer term our data will allow to build better mechanistic and predictive models of glass formation. This will require comprehensive theoretical studies, and this will be supported by a larger database of FIM data determined for >10 additional alloy systems. This is obviously far beyond the scope of this manuscript. But our manuscript lays the foundation and inspires a renewed research thrust in this direction.

For the first point, the authors claimed that-*“We find that the fragility varies over a remarkably wide range in Mg-Cu-Y. Large fragility variations have commonly been attributed to different materials classes, due to differences in interaction and structure type. More recent evidence suggests that large fragility variations may also occur amongst metallic glasses, even within the same alloy system. Here, in Mg-Cu-Y we observe the largest change of fragility with composition*

that has been reported within a metallic glass forming system to date." As I had mentioned before (also remained by the reviewer 2), large variations have been reported in other glass forming liquids. Maybe the variation rate is relatively larger in Mg-Cu-Y system, however, the underlying mechanism remains unclear. It cannot be an important new insight into the fragility.

It is clear that the mechanism underlying the large fragility variations must rely on changes of the atomic and electronic structure in the liquid. Elucidating in detail the complex mechanisms by which the structure determines the liquid properties is a grand challenge, which is comparable to unraveling the microstructure-property relationships in crystalline solids. This challenge will likely be the focal point of liquid state physics for the next 20 years.

Our research cannot accomplish within a single manuscript what the entire research community has been working to understand for decades. However, our manuscript provides novel insights and introduces a new method. We want to remind the reviewer that just like revealing microstructure-property relationships, unraveling the chemical composition-fragility relationships is a grand challenge, which cannot be solved even with a single disruptive step forward.

Of course, we do agree with the reviewer that large fragility variations have previously been observed. In fact, we make the same argument in the manuscript:

More recent evidence suggests that large fragility variations may also occur amongst metallic glasses, even within the same alloy system ^{59, 60, 61}.

However, the changes of fragility with composition observed here within a single alloys system are larger than any other previously reported. We refer the reviewer to the above findings summary as well as to our below response and our previous review response (page 27 and following), where we have already explained in great detail why this finding is significant in the context with the other observations made.

For the second point, the authors claimed that-*"We find that a low fragility does not necessarily correlate with a high glass forming ability. The established models which are based on this correlation overestimate the contribution of the fragility because they are incomplete."* This conclusion is still not entirely new. The poor correlation between the fragility and the GFA is well-known.

We already addressed this point in detail in the first review response (page 28) and reiterate our response here:

Of course, we agree with the reviewer that it is widely known that the fragility does not determine GFA alone. Indeed, prominent examples exist, which show that a high GFA is possible despite a high fragility, and vice versa:

While intuitively reasonable, examples which do not follow this correlation have been reported^{17, 22, 39, 40}. Most notably, $Pd_{42.5}Cu_{30}Ni_{7.5}P_{20}$ is considered the glass former with the highest known critical casting thickness and equivalently the lowest critical cooling rate, but it exhibits a comparatively high fragility of $m = 58$ ^{17, 22}. Conversely, $Zr_{80}Pt_{20}$ appears to exhibit a low fragility, but does not form a glass even in melt-spun ribbons³⁹.

Nonetheless, it is understood that a lower fragility generally leads to higher GFA.

Most importantly, it has been widely suggested that strong liquids are correlated with high GFA^{14, 17, 18, 19, 21, 29, 30, 33, 34, 35, 36, 37, 38}. Strong liquids exhibit higher viscosities across the vitrification temperature range, which slows the crystallization kinetics and thus reduces the critical cooling rate R_c required for glass formation. In addition, strong liquids are thought to be more densely packed and hence exhibit an enthalpy closer to the competing crystal phases, which reduces the driving force of crystallization²¹.

Based on this, we would initially expect to see some degree of correlation between a low fragility and high GFA, in particular given that we found the largest currently reported variation of fragility with composition. Yet, we do not see such a correlation.

The above is an important observation which acts as the starting point for our following analysis. But we want to emphasize to the reviewer once again: We know and clearly say in the manuscript that it is already understood that the fragility alone does not determine GFA. We do not claim that this is a new finding of our study.

By contrast, the new conclusion we do reach takes the above observation a step further:

Our data allow us to reveal that the state-of-the-art models are incomplete. Those models use a combination of T_{rg} and m . They state that the combination of a large T_{rg} and a small m lead to high GFA. Our conclusion is that these models, even though they do account for those **two contributions** rather than just the fragility alone, are **still incomplete** and overestimate the contribution of the fragility.

Our above statement cited by the reviewer is not part of the manuscript. We wrote those sentences only in the review response (page 27). We acknowledge that this statement may be confusing because “based on this correlation” sounds as though we were talking about models

based exclusively on the fragility. But that is not what we mean and in our first review response we clarified this matter in much more detail on page 28. We invite the reviewer to look at page 28 of our first review response again.

Further, in the revised manuscript we explained our conclusion clearly:

Overall, the above observations [here we are referring to the finding that the established predictive GFA models based on both T_{rg} and m still perform poorly across Mg-Cu-Y] suggest that the fragility is not the dominant contribution to GFA in Mg-Cu-Y. The quantitative models^{17, 18} seem to overestimate the contribution of m .

This leads to the conclusion that additional features must be required to model GFA for the broader spectrum of metallic glasses. The established models are incomplete:

We argue that Johnson's model is effective when applied to bulk metallic glasses with high GFA¹⁷. However, T_{rg} and m alone are insufficient to explain GFA over the full GFA spectrum. Our Mg-Cu-Y data cover a broad composition region, in which GFA ranges from poor glass formers to BMG formers. Thus, additional features can emerge as decisive contributions to GFA.

Since they have all the data of GFA, m , T_{rg} , t^*_x etc. as shown in Fig.4, why not plot T_{rg} , m or t^*_x with GFA to provide direct correlation? Therefore, we can see more intuitively which parameters are good and which parameters are not good.

We already explained above that the color maps in Fig. 4 provide the best visualization of our data. They allow the reader to easily compare quantities and identify correlations or discrepancies.

Additional reasons why it is not good to show scatter plots between the different parameters:

a) The quantities in Fig. 4 are complex functions of the composition. It is essential to show this compositional dependence and allow the reader to see in what compositional regions parameters do or do not correlate well. In scatter plots the compositional information is completely lost.

b) It is not helpful, but rather confusing and distracting, to use scatterplots to show correlations between one parameter and another when a parameter actually depends on more than one other parameters. For instance, GFA is a function of at least T_{rg} and m . Thus, a scatterplot of GFA with only T_{rg} or only m would not be insightful. The scatterplot between GFA and m would only

show that the two are not well correlated. But the colormaps in Fig. 4 already show this much more clearly, because they incorporate the full compositional information. Similarly, Fig. 4c shows clearly and intuitively that T_{rg} correlates strongly with GFA at all compositions.

In this review response we show one such example scatter plot on page 13. This example illustrates our above arguments. The scatter plot does not provide a useful visualization to the reader. In the scatterplot information is lost, the plot is confusing, and it would clutter the manuscript, which is currently very concise and effective in communicating our story.

By contrast, when comparing between the colormaps there is no barrier whatsoever against seeing whether a correlation is good or poor. The colormaps present the full level of detail of the data, and at the same time allow the reader to easily compare between parameters.

For the third point, the authors claimed that-*“We identify crystallization complexity as additional contribution to GFA, which becomes decisive when comparing alloys across orders of magnitude of GFA. To quantify this contribution, we introduce the $\hat{\tau}_x$ indicator.”*

First of all, we still do not know how good the parameter $\hat{\tau}_x$ is.

The first three paragraphs of section 6 provide a solid description that crystallization complexity is an important contribution to glass forming ability which was not accounted for in previous models.

The question now becomes how this contribution can be quantified. We propose the parameter $\hat{\tau}_x$ and explain why this parameter allows to estimate crystallization complexity. Our data in Fig. 4e show a strong correlation with GFA, which illustrates and supports our argument.

Altogether, $\hat{\tau}_x$ is a promising approach and in future work the community will evaluate how successful and widely applicable this parameter is.

Second, I agree with reviewer 1 that the related part is half-baked. Only if the authors are able to relate $\hat{\tau}_x$ to m based on the data in Fig. 2c3 and the definition of $\hat{\tau}_x$, they can claim to have reached an important new insights into the fragility and its contribution to glass formation. If not, how do they evaluate the contribution of m to the GFA? The data in Fig. S7 should be important to extract some hints related to this issue.

We already addressed the feedback by reviewer #1 extensively in the first review response and made revisions to the manuscript which significantly improved this discussion. Since reviewer #1 is satisfied with our revisions, it is unclear what reviewer #3 thinks is missing beyond the revisions already made.

Secondly, we do relate \hat{t}_x to GFA, which is the goal of this parameter:

Figure 4e shows that \hat{t}_x varies widely and correlates with GFA (details in SI).

Thirdly, it is not clear why the reviewer asks us to relate \hat{t}_x and m . Based on the current evidence, these two parameters are independent. In fact, the essence of our definition of \hat{t}_x is that we want to isolate crystallization complexity as an independent parameter, in which we eliminate the fragility dependence of raw crystallization times.

Indeed, we had already addressed this point extensively in our revised manuscript: Reviewer #2 had requested a visualization to show how our parameter \hat{t}_x is better than the bare crystallization time t_x . Thus, in the revised manuscript we incorporated Supplementary Figure 7, which is displayed again below and which clearly reveals that the bare crystallization time is significantly affected by the fragility and therefore not an independently meaningful contribution to GFA. By contrast, this fragility dependence is effectively eliminated through our definition of \hat{t}_x , making \hat{t}_x an independently meaningful indicator of crystallization complexity, which exhibits a strong correlation with GFA. Altogether, this provides ample support of our crystallization complexity idea and the \hat{t}_x parameter. Both reviewers #1 and #2 found these data and arguments convincing.

Comparison between \hat{t}_x and $t_x(\dot{T} \approx 25 \text{ K/min})$: \hat{t}_x allows to eliminate the alloy-specific temperature dependence of crystallization kinetics. To highlight the improved perspective this offers, we compare \hat{t}_x to $t_x(\dot{T} \approx 25 \text{ K/min})$ in Supplementary Figure 7a. The latter is the time to crystallization under heating conditions. It is determined from our FIM curves as $t'_{\tau=200 \text{ sec}} - t'_x$, i.e. as the difference between the bounds of integration used in Supplementary Equation (9).

For comparison we highlight two regions: The region around point ① exhibits high GFA, but the observed crystallization time of only ~ 100 seconds is comparatively low. Meanwhile, the region around point ② exhibits much lower GFA, but the crystallization time is about twice as long at ~ 200 seconds. These crystallization times seem to contradict the observed variation of GFA. This is misleading, however, because the crystallization time alone is meaningless if the underlying kinetic rates and their temperature dependence are not considered. Indeed, in region ① the fragility is significantly higher. Accordingly, η_x , the viscosity at the point of crystallization, is an order of magnitude lower here, the associated kinetics are ten times faster, and the crystallization time is shorter. By contrast, the independent \hat{t}_x indicator accounts for such temperature dependence, reveals that the crystallization complexity is actually higher in region ①, and strongly correlates with GFA.

Supplementary Figure 7: (a) Crystallization time t_x under heating conditions of ~ 25 K/min, compared to (b) \hat{t}_x the crystallization time in relaxation time units. For reference, (c) shows the fragility m and (d) shows η_x .

The above colormaps are very effective at communicating why \hat{t}_x is a useful and how it relates to other parameters. By contrast, scatter plots as suggested by reviewer #3 are not useful to visualize such complex relationships.

For illustration, we show the scatter plot between \hat{t}_x and m below, as suggested by reviewer #3. In such a scatter plot the compositional information, which is key to interpreting these data properly, is lost entirely. Accordingly, the key insights into the relationships between parameters are also lost, and no obvious correlations or discrepancies can be identified. Only by comparing the colormaps can the reader easily grasp how the parameters correlate or differ as a function of composition. The colormaps make these complex relationships accessible.

In conclusion, we would like to highlight once again that crystallization complexity is a decisive contribution to GFA, but it has not been considered in previous models. \hat{t}_x offers an effective way to estimate this contribution quantitatively. Data from many more alloy systems will be required to show quantitatively how m , T_{rg} , and \hat{t}_x exactly determine GFA together. Within this manuscript, \hat{t}_x is a novel idea of importance.

This work only draws some rough conclusions at each point and does not touch on the nature behind even one phenomenon or establish one quantitative correlation. SO, I don't think it meets the requirements of Nature communications.

We hope that we were able to clarify our arguments. Once again, we invite the reviewer to revisit our manuscript and recognize the many merits of our work.

REVIEWER COMMENTS

Reviewer #1 (Remarks to the Author):

The authors adequately addressed the concerns described earlier and significantly modified the text. I recommend publication as revised.

Reviewer #2 (Remarks to the Author): 
Please see the attached file.

Reviewer #3 (Remarks to the Author): 
Please see the attached file.

Reviewer Response, Submission NCOMMS-22-00021A (second revisions)

“Compositional dependence of the fragility in metallic glass forming liquids”

Sebastian A. Kube and Jan Schroers, et al.

We thank the reviewers for carefully reading the manuscript, and for providing constructive comments and suggestions to improve the manuscript. Based on these comments, we have fundamentally revised the relevant sections.

*Please find below in **blue** our detailed, point-by-point responses to the reviewers’ comments. Also, we provide supporting segments from the revised manuscript in **light blue italic**. All changes have been highlighted in blue font in the revised manuscript.*

Reviewer #1:

The authors adequately addressed the concerns described earlier and significantly modified the text. I recommend publication as revised.

We thank the reviewer for this assessment.

Reviewer #2:

As I stated in my earlier report, this is an extensive investigation that produced a huge amount of experimental data which will be useful for researchers in this field. There were a few open questions that myself and other referees raised. The authors have tried to answer those questions as far as possible and revised the manuscript accordingly. Aside from the importance of the new data, the authors have proposed a new metric to assess glass formability. Although not an open and shut case, the new metric may, to some extent, quantify the complexity of crystal phase/phases formation. It is understood by many active researchers in this field that complex pathways from glass/supercooled liquid to crystal are important factors for glass formation; whether that complexity comes from the structural difference between the liquid and crystal phases is an open question. I am not aware of any previous work that tried to quantify this complexity. As a first attempt in this direction this work is important and merits publication. Time will tell how useful this metric will be!

We thank the reviewer for this assessment and agree.

The raw data that came out from this investigation will be of interest to researchers who are trying to develop better theories for glass formation. The authors have provided some links in the supplement sections where such data will be available. I think they should also provide such data for the fragility parameter, in case I missed such a link.

We thank the reviewer for pointing this out. We agree that the reader should be able to find these valuable data easily. Indeed, the fragility data are already enclosed in the "SupplementaryFiles_ResearchData" folder, which we uploaded upon submission. To help the reader to quickly locate the fragility data, we have added the following statement to the Supplementary Contents section as recommended by the reviewer:

All FIM data are enclosed in "FIMData.xlsx", including the fragility data.

Reviewer #3:

I had carefully read the revised manuscript and the response to the comments. Some improvement can be found, but there is no substantial improvement.

We thank the reviewer for the time and effort taken to assess our revised manuscript.

However, we are disappointed by the reviewer's assessment as it shares neither our own excitement for this research, nor the support and approval we receive from the other two reviewers, nor the enthusiastic interest we have received from the metallic glass community when we presented these results at two recent conference meetings and two seminars.

We want to reiterate that we have already addressed all of the reviewer's comments in great detail in the first review round. It appears that the reviewer is actually satisfied with the majority of these revisions, as these aspects are no longer mentioned in this second assessment.

In particular the resolved aspects we addressed include:

- Evaluating the predictability of literature GFA models.
- Justification of Figure 1.
- Connection between fragility and structure
- Why we decided against using machine learning here.

Since these comments have been resolved, one can conclude that the manuscript has in fact improved substantially.

In the first review round we also clarified why our findings are novel and of significant interest to the community. The other two reviewers are also excited about these findings. Yet, reviewer #3 does still not seem to be satisfied. We will try to address this through additional clarification.

The reason why I think this paper is a result based on phenomenon description is that it does not give any characterization of the correlation between parameters.

Our manuscript provides extensive quantitative information on T_g , T_x , T_L , GFA, m , T_{rg} , d_{calc} , \hat{t}_x , and η_x . Each quantity is visualized as a function of composition over a wide range. This is a comprehensive quantitative assessment. Thanks to our FIM data, this assessment goes far beyond previous published analyses.

To address the reviewer's concerns "that no characterization of the correlation between parameters is given", we emphasize below that an extensive characterization of such correlations has in fact been carried out. Most importantly:

a) We characterize the correlation between m and GFA:

When comparing the fragility and GFA maps (Figure 4a and b), the area of highest GFA coincides with the ridge of highest fragility values. This is remarkable, since according to the current understanding the opposite should be the case; high GFA should be associated with a low fragility 14, 17, 18, 19, 21, 29, 30, 33, 34, 35, 36, 37, 38. Evidently, this correlation does not have to be strictly fulfilled.

b) We characterize the correlation between T_{rg} and GFA:

We find that the region of high T_{rg} coincides well with the region of high GFA. This is primarily due to a deep eutectic located at $Mg_{65}Cu_{25}Y_{10}$ ⁵⁶. From this perspective, T_{rg} appears to be the dominant contribution to GFA within this alloy system.

c) We evaluate the performance of d_{calc} , and thereby characterize how the previously established model combination of m and T_{rg} correlates with GFA:

At $Mg_{65}Cu_{25}Y_{10}$ we find approximate agreement (here: 11.6 mm, Johnson 17: 8.8 mm, experimental 55: 7 mm), suggesting potentially useful GFA predictions. However, over the wider composition range the agreement is not good. Even far from the bulk forming region, d_{calc} values of 5 mm or higher are predicted. This discrepancy is particularly obvious on the boundary line of crystallization under sputtering conditions. Here, the critical cooling rate is $\sim 10^8$ K/s, which corresponds to a critical casting thickness of $< 1 \mu m$ ^{26, 70}. Yet, for $Mg_{72}Cu_{19}Y_9$ ($m = 31.8$, $T_{rg} = 0.53$) a d_{calc} value of 7 mm is predicted, which is four orders of magnitude larger. This discrepancy cannot result from hypothetical errors in the determined fragility since Eq. (6) would require $m = 190$ for $1 \mu m$.

The qualitative agreement is also poor. d_{calc} does not match the narrow, long contour of the high GFA region. Further, at $Mg_{60}Cu_{30}Y_{10}$ a GFA as high as $d_{calc} = 25$ mm is predicted, resulting from low fragility values $m \approx 25$ overlapping with high T_{rg} values ≥ 0.56 . The experimental GFA data do not support this prediction.

d) We characterize the correlation between \hat{t}_x and GFA:

Error! Reference source not found.e shows that \hat{t}_x varies widely and correlates with GFA (details in SI).

e) We characterize the correlation between \hat{t}_x and η_x :

A higher crystallization complexity allows to reach lower η_x , even for those compositions which exhibit the highest fragility values.

f) We show and describe many additional correlations throughout the manuscript and in the Supplementary Information.

We can't get any quantitative information from Fig 4, but can only roughly judge whether the correlation is good or bad through the eyes.

We respectfully disagree.

Every single panel in Figure 4 provides extensive quantitative information: The position of each marker unambiguously encodes the composition of the corresponding alloy. The color of each marker unambiguously encodes the value of the corresponding quantity. The colorbars link the marker color to the quantity value. Thus, each of these colormaps provides quantitative information throughout.

The colormaps in Fig. 4 provide a strong visual representation of our data. They allow the reader to easily grasp and compare the composition dependency of each quantity. The reader can easily recognize correlations and discrepancies between all quantities.

I agree that this work provides a useful new method for measuring dynamic behaviors of glass forming liquids. Based on this experimental technique, the authors measured a number of data within one ternary alloy system, but it seems to me that this is a waste of the value of these data and thus loses the point of high-throughput research.

We respectfully disagree with the reviewer's statement. The value of these data is not wasted. Within this manuscript, these data unfold value in three ways:

- 1) Firstly, these data clearly demonstrate the robustness of our new FIM method.
- 2) Secondly, we have gleaned a number of significant insights from these data. These are:
 - The fragility varies over a remarkably wide range in Mg-Cu-Y. Large fragility variations have commonly been attributed to different materials classes, due to the fundamental differences in interaction and structure type. More recent evidence suggests that large fragility variations may also occur amongst metallic glasses, even within the same alloy system. Here, in Mg-Cu-Y we observe the largest change of fragility with composition that has been reported within a metallic glass forming system to date. The observed fragility variations must originate primarily from composition-dependent changes to the atomic and electronic structure, which presumably lead to different packing motifs at different compositions and altered atomic interaction potentials.
 - We find that a low fragility does not necessarily correlate with a high glass forming ability. Moreover, even the latest established and widely accepted literature models, which are based on the two correlations that a small fragility and a large T_{rg} lead to good GFA, clearly overestimate the contribution of the fragility in Mg-Cu-Y.

- Our data are the first to represent the wider spectrum of metallic glass forming alloys. Under these circumstances the previous models do not successfully predict GFA and thus do not offer useful predictions across Mg-Cu-Y. We find that these models are incomplete. Improved models must incorporate additional contributions to represent the broad spectrum of metallic glass forming alloys.
 - We identify the crystallization complexity as one such contribution. Our data suggest that this contribution becomes decisive when comparing alloys across many orders of magnitude of GFA.
 - To quantify this contribution for the first time, we propose the \hat{t}_x indicator. This indicator offers an approximation which in the long term will certainly be assessed and refined by the community. Within the manuscript, we explain that this parameter offers very reasonable and insightful estimates which capture the essence of crystallization complexity. Our data correlate well with GFA and support this proposal.
- 3) Thirdly, the above insights form the foundation to incorporate the concept of crystallization complexity into models of glass formation. The data show that the previous understanding of fragility and glass forming ability is insufficient and this additional contribution of crystallization complexity must be considered.

For the wider metallic glass community these data also unfold value: We provide all data in a clean format to the reader. This means that these data will be permanently and openly available to the entire community. We anticipate that other researchers will use these data to identify and test new correlations and build better models. This is most certainly in the spirit of modern high-throughput research.

We believe, and reviewers #1 and #2 share this view, that in the longer term our data will allow to build better mechanistic and predictive models of glass formation. This will require comprehensive theoretical studies, and this will be supported by a larger database of FIM data determined for >10 additional alloy systems. This is obviously far beyond the scope of this manuscript. But our manuscript lays the foundation and inspires a renewed research thrust in this direction.

For the first point, the authors claimed that-*“We find that the fragility varies over a remarkably wide range in Mg-Cu-Y. Large fragility variations have commonly been attributed to different materials classes, due to differences in interaction and structure type. More recent evidence suggests that large fragility variations may also occur amongst metallic glasses, even within the same alloy system. Here, in Mg-Cu-Y we observe the largest change of fragility with composition*

that has been reported within a metallic glass forming system to date." As I had mentioned before (also remained by the reviewer 2), large variations have been reported in other glass forming liquids. Maybe the variation rate is relatively larger in Mg-Cu-Y system, however, the underlying mechanism remains unclear. It cannot be an important new insight into the fragility.

It is clear that the mechanism underlying the large fragility variations must rely on changes of the atomic and electronic structure in the liquid. Elucidating in detail the complex mechanisms by which the structure determines the liquid properties is a grand challenge, which is comparable to unraveling the microstructure-property relationships in crystalline solids. This challenge will likely be the focal point of liquid state physics for the next 20 years.

Our research cannot accomplish within a single manuscript what the entire research community has been working to understand for decades. However, our manuscript provides novel insights and introduces a new method. We want to remind the reviewer that just like revealing microstructure-property relationships, unraveling the chemical composition-fragility relationships is a grand challenge, which cannot be solved even with a single disruptive step forward.

Of course, we do agree with the reviewer that large fragility variations have previously been observed. In fact, we make the same argument in the manuscript:

More recent evidence suggests that large fragility variations may also occur amongst metallic glasses, even within the same alloy system^{59, 60, 61}.

However, the changes of fragility with composition observed here within a single alloys system are larger than any other previously reported. We refer the reviewer to the above findings summary as well as to our below response and our previous review response (page 27 and following), where we have already explained in great detail why this finding is significant in the context with the other observations made.

For the second point, the authors claimed that-*"We find that a low fragility does not necessarily correlate with a high glass forming ability. The established models which are based on this correlation overestimate the contribution of the fragility because they are incomplete."* This conclusion is still not entirely new. The poor correlation between the fragility and the GFA is well-known.

We already addressed this point in detail in the first review response (page 28) and reiterate our response here:

Of course, we agree with the reviewer that it is widely known that the fragility does not determine GFA alone. Indeed, prominent examples exist, which show that a high GFA is possible despite a high fragility, and vice versa:

While intuitively reasonable, examples which do not follow this correlation have been reported^{17, 22, 39, 40}. Most notably, $Pd_{42.5}Cu_{30}Ni_{7.5}P_{20}$ is considered the glass former with the highest known critical casting thickness and equivalently the lowest critical cooling rate, but it exhibits a comparatively high fragility of $m = 58$ ^{17, 22}. Conversely, $Zr_{80}Pt_{20}$ appears to exhibit a low fragility, but does not form a glass even in melt-spun ribbons³⁹.

Nonetheless, it is understood that a lower fragility generally leads to higher GFA.

Most importantly, it has been widely suggested that strong liquids are correlated with high GFA^{14, 17, 18, 19, 21, 29, 30, 33, 34, 35, 36, 37, 38}. Strong liquids exhibit higher viscosities across the vitrification temperature range, which slows the crystallization kinetics and thus reduces the critical cooling rate R_c required for glass formation. In addition, strong liquids are thought to be more densely packed and hence exhibit an enthalpy closer to the competing crystal phases, which reduces the driving force of crystallization²¹.

Based on this, we would initially expect to see some degree of correlation between a low fragility and high GFA, in particular given that we found the largest currently reported variation of fragility with composition. Yet, we do not see such a correlation.

The above is an important observation which acts as the starting point for our following analysis. But we want to emphasize to the reviewer once again: We know and clearly say in the manuscript that it is already understood that the fragility alone does not determine GFA. We do not claim that this is a new finding of our study.

By contrast, the new conclusion we do reach takes the above observation a step further:

Our data allow us to reveal that the state-of-the-art models are incomplete. Those models use a combination of T_{rg} and m . They state that the combination of a large T_{rg} and a small m lead to high GFA. Our conclusion is that these models, even though they do account for those **two contributions** rather than just the fragility alone, are **still incomplete** and overestimate the contribution of the fragility.

Our above statement cited by the reviewer is not part of the manuscript. We wrote those sentences only in the review response (page 27). We acknowledge that this statement may be confusing because “based on this correlation” sounds as though we were talking about models

based exclusively on the fragility. But that is not what we mean and in our first review response we clarified this matter in much more detail on page 28. We invite the reviewer to look at page 28 of our first review response again.

Further, in the revised manuscript we explained our conclusion clearly:

Overall, the above observations [here we are referring to the finding that the established predictive GFA models based on both T_{rg} and m still perform poorly across Mg-Cu-Y] suggest that the fragility is not the dominant contribution to GFA in Mg-Cu-Y. The quantitative models^{17, 18} seem to overestimate the contribution of m .

This leads to the conclusion that additional features must be required to model GFA for the broader spectrum of metallic glasses. The established models are incomplete:

We argue that Johnson's model is effective when applied to bulk metallic glasses with high GFA¹⁷. However, T_{rg} and m alone are insufficient to explain GFA over the full GFA spectrum. Our Mg-Cu-Y data cover a broad composition region, in which GFA ranges from poor glass formers to BMG formers. Thus, additional features can emerge as decisive contributions to GFA.

Since they have all the data of GFA, m , T_{rg} , t^*_x etc. as shown in Fig.4, why not plot T_{rg} , m or t^*_x with GFA to provide direct correlation? Therefore, we can see more intuitively which parameters are good and which parameters are not good.

We already explained above that the color maps in Fig. 4 provide the best visualization of our data. They allow the reader to easily compare quantities and identify correlations or discrepancies.

Additional reasons why it is not good to show scatter plots between the different parameters:

a) The quantities in Fig. 4 are complex functions of the composition. It is essential to show this compositional dependence and allow the reader to see in what compositional regions parameters do or do not correlate well. In scatter plots the compositional information is completely lost.

b) It is not helpful, but rather confusing and distracting, to use scatterplots to show correlations between one parameter and another when a parameter actually depends on more than one other parameters. For instance, GFA is a function of at least T_{rg} and m . Thus, a scatterplot of GFA with only T_{rg} or only m would not be insightful. The scatterplot between GFA and m would only

show that the two are not well correlated. But the colormaps in Fig. 4 already show this much more clearly, because they incorporate the full compositional information. Similarly, Fig. 4c shows clearly and intuitively that T_{rg} correlates strongly with GFA at all compositions.

In this review response we show one such example scatter plot on page 13. This example illustrates our above arguments. The scatter plot does not provide a useful visualization to the reader. In the scatterplot information is lost, the plot is confusing, and it would clutter the manuscript, which is currently very concise and effective in communicating our story.

By contrast, when comparing between the colormaps there is no barrier whatsoever against seeing whether a correlation is good or poor. The colormaps present the full level of detail of the data, and at the same time allow the reader to easily compare between parameters.

For the third point, the authors claimed that-*“We identify crystallization complexity as additional contribution to GFA, which becomes decisive when comparing alloys across orders of magnitude of GFA. To quantify this contribution, we introduce the $\hat{\tau}_x$ indicator.”*

First of all, we still do not know how good the parameter $\hat{\tau}_x$ is.

The first three paragraphs of section 6 provide a solid description that crystallization complexity is an important contribution to glass forming ability which was not accounted for in previous models.

The question now becomes how this contribution can be quantified. We propose the parameter $\hat{\tau}_x$ and explain why this parameter allows to estimate crystallization complexity. Our data in Fig. 4e show a strong correlation with GFA, which illustrates and supports our argument.

Altogether, $\hat{\tau}_x$ is a promising approach and in future work the community will evaluate how successful and widely applicable this parameter is.

Second, I agree with reviewer 1 that the related part is half-baked. Only if the authors are able to relate $\hat{\tau}_x$ to m based on the data in Fig. 2c3 and the definition of $\hat{\tau}_x$, they can claim to have reached an important new insights into the fragility and its contribution to glass formation. If not, how do they evaluate the contribution of m to the GFA? The data in Fig. S7 should be important to extract some hints related to this issue.

We already addressed the feedback by reviewer #1 extensively in the first review response and made revisions to the manuscript which significantly improved this discussion. Since reviewer #1 is satisfied with our revisions, it is unclear what reviewer #3 thinks is missing beyond the revisions already made.

Secondly, we do relate \hat{t}_x to GFA, which is the goal of this parameter:

Figure 4e shows that \hat{t}_x varies widely and correlates with GFA (details in SI).

Thirdly, it is not clear why the reviewer asks us to relate \hat{t}_x and m . Based on the current evidence, these two parameters are independent. In fact, the essence of our definition of \hat{t}_x is that we want to isolate crystallization complexity as an independent parameter, in which we eliminate the fragility dependence of raw crystallization times.

Indeed, we had already addressed this point extensively in our revised manuscript: Reviewer #2 had requested a visualization to show how our parameter \hat{t}_x is better than the bare crystallization time t_x . Thus, in the revised manuscript we incorporated Supplementary Figure 7, which is displayed again below and which clearly reveals that the bare crystallization time is significantly affected by the fragility and therefore not an independently meaningful contribution to GFA. By contrast, this fragility dependence is effectively eliminated through our definition of \hat{t}_x , making \hat{t}_x an independently meaningful indicator of crystallization complexity, which exhibits a strong correlation with GFA. Altogether, this provides ample support of our crystallization complexity idea and the \hat{t}_x parameter. Both reviewers #1 and #2 found these data and arguments convincing.

Comparison between \hat{t}_x and $t_x(\dot{T} \approx 25 \text{ K/min})$: \hat{t}_x allows to eliminate the alloy-specific temperature dependence of crystallization kinetics. To highlight the improved perspective this offers, we compare \hat{t}_x to $t_x(\dot{T} \approx 25 \text{ K/min})$ in Supplementary Figure 7a. The latter is the time to crystallization under heating conditions. It is determined from our FIM curves as $t'_{\tau=200 \text{ sec}} - t'_x$, i.e. as the difference between the bounds of integration used in Supplementary Equation (9).

For comparison we highlight two regions: The region around point ① exhibits high GFA, but the observed crystallization time of only ~ 100 seconds is comparatively low. Meanwhile, the region around point ② exhibits much lower GFA, but the crystallization time is about twice as long at ~ 200 seconds. These crystallization times seem to contradict the observed variation of GFA. This is misleading, however, because the crystallization time alone is meaningless if the underlying kinetic rates and their temperature dependence are not considered. Indeed, in region ① the fragility is significantly higher. Accordingly, η_x , the viscosity at the point of crystallization, is an order of magnitude lower here, the associated kinetics are ten times faster, and the crystallization time is shorter. By contrast, the independent \hat{t}_x indicator accounts for such temperature dependence, reveals that the crystallization complexity is actually higher in region ①, and strongly correlates with GFA.

Supplementary Figure 7: (a) Crystallization time t_x under heating conditions of ~ 25 K/min, compared to (b) \hat{t}_x the crystallization time in relaxation time units. For reference, (c) shows the fragility m and (d) shows η_x .

The above colormaps are very effective at communicating why \hat{t}_x is a useful and how it relates to other parameters. By contrast, scatter plots as suggested by reviewer #3 are not useful to visualize such complex relationships.

For illustration, we show the scatter plot between \hat{t}_x and m below, as suggested by reviewer #3. In such a scatter plot the compositional information, which is key to interpreting these data properly, is lost entirely. Accordingly, the key insights into the relationships between parameters are also lost, and no obvious correlations or discrepancies can be identified. Only by comparing the colormaps can the reader easily grasp how the parameters correlate or differ as a function of composition. The colormaps make these complex relationships accessible.

In conclusion, we would like to highlight once again that crystallization complexity is a decisive contribution to GFA, but it has not been considered in previous models. \hat{t}_x offers an effective way to estimate this contribution quantitatively. Data from many more alloy systems will be required to show quantitatively how m , T_{rg} , and \hat{t}_x exactly determine GFA together. Within this manuscript, \hat{t}_x is a novel idea of importance.

This work only draws some rough conclusions at each point and does not touch on the nature behind even one phenomenon or establish one quantitative correlation. SO, I don't think it meets the requirements of Nature communications.

We hope that we were able to clarify our arguments. Once again, we invite the reviewer to revisit our manuscript and recognize the many merits of our work.